# A *rad50* germline mutation induces tumorigenesis and ataxia-telangiectasia phenotype in a transparent medaka model

Shinichi Chisada[1☯], Kouki Ohtsuka[2☯]*, Masachika Fujiwara[3], Masao Yoshida[1], Satsuki Matsushima[2], Takashi Watanabe[2], Kanae Karita[1], Hiroaki Ohnishi[2]

1 Department of Hygiene and Public Health, Kyorin University School of Medicine, Mitaka, Tokyo, Japan,
2 Department of Laboratory Medicine, Kyorin University School of Medicine, Mitaka, Tokyo, Japan,
3 Department of Pathology, Kyorin University School of Medicine, Mitaka, Tokyo, Japan

☯ These authors contributed equally to this work.
* kouki7@ks.kyorin-u.ac.jp

**Data Availability Statement:** All data are within the paper and its Supporting Information files.

## Abstract

The MRE11A-RAD50-NBS1 complex activates the ataxia-telangiectasia mutated (ATM) pathway and plays a central role in genome homeostasis. The association of *RAD50* mutations with disease remains unclear; hence, we adopted a medaka *rad50* mutant to demonstrate the significance of *RAD50* mutation in pathogenesis using the medaka as an experimental animal. A 2-base pair deletion in the *rad50* gene was introduced into transparent STIII medaka using the CRISPR/Cas9 system. The mutant was analyzed histologically for tumorigenicity and hindbrain quality, as well as for swimming behavior, to compare with existing *ATM-*, *MRE11A-*, and *NBS1*-mutation-related pathology. Our results revealed that the medaka *rad50* mutation concurrently reproduced tumorigenesis (8 out of 10 $rad50^{\Delta2/+}$ medaka), had a decrease in median survival time (65.7 ± 1.1 weeks in control vs. 54.2 ± 2.6 weeks in $rad50^{\Delta2/+}$ medaka, $p = 0.001$, Welch's *t*-test), exhibited semi-lethality in $rad50^{\Delta2/\Delta2}$ medaka and most of the major ataxia-telangiectasia phenotypes, including ataxia (rheotaxis ability was lower in $rad50^{\Delta2/+}$ medaka than in the control, Mann–Whitney U test, $p < 0.05$), and telangiectasia (6 out of 10 $rad50^{\Delta2/+}$ medaka). The fish model may aid in further understanding the tumorigenesis and phenotype of ataxia-telangiectasia-related *RAD50* germline mutations and in developing novel therapeutic strategies against RAD50 molecular disorders.

## Introduction

RAD50 is a molecule that, along with meiotic recombination 11 (MRE11) and Nijmegen breakage syndrome 1 (NBS1), constitutes the MRE11-RAD50-NBS1 (MRN) complex [1]. The MRN complex is conserved in all eukaryotes and plays a central role in maintaining genome homeostasis, particularly by repairing DNA double-strand breaks (DSBs) [2,3] that occur during meiosis, homologous recombination, non-homologous end joining, DNA damage [4], and telomere management [5]. This complex activates ataxia-telangiectasia mutated (ATM) kinase [6], which initiates mechanisms associated with DNA repair, cell cycle arrest, and apoptosis [7].

**Funding:** This study was supported by JSPS KAKENHI (grant number JP20K08992). The funders had no role in study design, data collection and analysis, decision to publish, or preparation of the manuscript.

Heterozygous and homozygous mutations in the *RAD50* gene have been reported in two patients with a NBS-like disorder (NBSLD [OMIM 613078]) phenotype, who exhibit microcephaly, growth retardation, and reduced ATM kinase signaling [8,9]; however, it is difficult to evaluate the pathophysiology of *RAD50* mutations in humans because these mutations are rare. Therefore, accumulation of patient information on a large scale is required to elucidate the health risks associated with *RAD50* mutations in humans. Furthermore, several studies using experimental animals have made it difficult to determine the biological effects of mutations in the *rad50* gene. For instance, early mouse embryos that are homozygous for mutated *rad50* alleles are lethal [10], which differs from an NBSLD patient with a homozygous mutation of the *RAD50* gene [9]. Mice with hypomorphic RAD50 present with hydrocephalus, defects in primitive hematopoietic and gametogenic cells, and liver tumorigenesis [11]. Moreover, this phenotype is not similar to that of NBSLD patients with a mutation in the *RAD50* gene [8]. Therefore, further validation is required to elucidate the significance of mutations in the *rad50* gene using experimental animals.

Mutations in *ATM*, *MRE11A*, and *NBS1* genes lead to similar pathogenesis in humans. Comparing these pathogeneses will aid in elucidating the significance of *RAD50* mutations. Additionally, these mutations have been recognized as a cause of hereditary and/or malignant diseases in humans. Germline mutations in *ATM* and *MRE11A* underlie the autosomal recessive ataxia-telangiectasia syndrome (A-T [OMIM 208900]) and ataxia-telangiectasia-like disorder 1 (ATLD1 [OMIM 604391]), respectively. A-T is characterized by ataxia, telangiectasia, immune defects, chromosomal breakage, predisposition to malignant tumors, and susceptibility to ionizing radiation (IR) [12,13]. Mice carrying homozygous *atm* mutations exhibit behavioral abnormalities, aberrations in cerebellar Purkinje neurons, and thymic lymphomas, but not telangiectasia [14]. Although patients with ATLD1 also display some similar signs to those with A-T (ataxia and susceptibility to IR), telangiectasia and immune deficiency are not present in these patients [15]. Homozygous *mre11a* mutations in mice result in embryonic lethality [16]. Germline homozygous mutations in *NBS1* in humans are responsible for NBS (OMIM 251260), which is characterized by microcephaly, growth retardation, immunodeficiency, predisposition to cancer, IR-sensitivity, and chromosomal instability in homozygous mutants [17]; in contrast, homozygous mutations in mice result in embryonic lethality [18].

Patients with A-T, ATLD1, and NBS show a predisposition to cancers, including breast cancer, bladder cancer, melanoma, lung cancer, colon cancer, leukemia, and lymphoma, thereby implying the tumorigenic effects of mutations in *ATM*, *MRE11A*, and *NBS1* genes. We recently identified a heterozygous *RAD50* I505fs*5 frameshift germline mutation in a 37-year-old man who presented with simultaneous duodenal and rectal cancers. Genomic analysis of the two cancers in the patient revealed a high tumor mutation burden and high microsatellite instability (details of the clinical and molecular features of the patient will be described elsewhere), suggesting that mutations in the *RAD50* gene predispose to cancers, as do mutations in *ATM*, *MRE11A*, and *NBS1* genes. To demonstrate whether the *RAD50* mutation identified in our patient contributed to the development of cancer, we generated and analyzed medaka with mutations in the corresponding exon of the *rad50* gene as that in the patient.

Medaka fish (*Oryzias latipes*) contain genes involved in basic DNA repair pathways in common with vertebrates; thus, they have often been used to detect toxic substances in water [19,20]. These fish have also been used as model organisms to study genes associated with carcinogenic mechanisms [21–23] and to elucidate pathways involved in genome homeostasis, such as the p53-mediated damage recognition pathway [24,25] and the base excision repair pathway [26,27]. Compared with mammalian models, fish models are advantageous because of their relatively short lifespans and rapid generation turnover. Previous studies have shown that zebrafish embryogenesis requires DNA repair mechanisms involving the MRN complex

and ATM kinases [28] and that medaka fish with *nbs1* gene polymorphisms collected from the wild population exhibit defective DSB repair abilities [29]. Therefore, the use of medaka models presents a promising approach to understanding the role of the MRN complex in DSB repair.

In this study, we aimed to 1) precisely understand the clinical significance of *RAD50* mutations and determine their role in tumorigenesis; 2) compare our mutant model with existing animal models. We successfully generated multiple tumors and typical A-T phenotypes with corresponding histological abnormalities, telangiectasia, immunological abnormalities, and ataxia in *rad50* mutant medaka using the CRISPR/Cas9 system. We also demonstrated that the mutant medaka had a shorter lifespan than the wildtype medaka.

## Materials and methods

### Fish and its husbandry

The medaka STIII strain used to generate medaka models of *rad50* deficiency was provided by the National BioResource Project Medaka (https://shigen.nig.ac.jp/medaka/). The STIII medaka fish is transparent because it is recessive to all pigments contained in the chromatophore; thus, it is also termed "transparent medaka." Its transparency allows the internal organs of live fish to be observed [30]. STIII does not show any organ defects other than the loss of pigmentation unless environmentally or genetically manipulated. Fish were reared under a 14 h:10 h light: dark photoperiod at 25–27°C in a tank with an aquatic filter (Tetra Auto One-Touch Filter AT-20, Spectrum Brands Japan Co. Ltd., Yokohama, Kanagawa, Japan). The water was filtered at a regular speed of 75 L/h, and water condition was maintained at pH 7.2–8.0, $NO^{3-} < 25$ mg/L, and dissolved oxygen concentration at 8.50–9.00 mg/L. This study was performed in accordance with the guidelines of the Animal Research Committee of Kyorin University (approval number 235 [for 2019–2021]).

### Determination of the nucleotide sequence of STIII medaka *rad50*

Nucleotide sequence data of human *RAD50* (ENSG00000113522) and medaka *rad50* (ENSORLG00000001678) were obtained from the Ensembl genome browser (http://www.ensembl.org/). The two sequences were aligned using the CLUSTAL W [31] server at DDBJ (clustalw.ddbj.nig.ac.jp/top-j.html) with default parameters. RAD50 has a coiled-coil structure that can be folded by the hook construct and carries MRE11- and NBS1-binding sites in its structure. The hook domain, a highly conserved homodimerization interface, performs an essential function in the MRN complex by recognizing damaged DNA and promoting DNA repair [1]. Thus, the sequence corresponding to the coiled-coil region before the hook construct of the medaka rad50 was estimated to be the putative exons 5–13. Next, the nucleotide sequences of intron 10, exon 11, intron 11, and exon 12 in *rad50* of STIII medaka were determined with an ABI Prism 3100-Avant genetic analyzer (Thermo Fisher Scientific, Waltham, MA, USA) using the primers shown in S1 Table.

### Mutant generation

The *RAD50* mutation identified in our patient with two cancers was a 2 bp deletion (c.1515_1516 del AA, p.I505fs*5) within exon 8 of the gene on chromosome 5q; exon 8 of human *RAD50* corresponds to the putative exon 11 of medaka *rad50*. The chromosomal location information of the medaka *rad50* is not shown in the Ensembl genome browser. Therefore, to generate the *rad50* mutant used in this study, guide RNAs; (5′-AGUUCAAAGCUCCAAUGUGG-3′) targeting exon 11 of *rad50* were designed using CCTop

([https://crispr.cos.uni-heidelberg.de/](https://crispr.cos.uni-heidelberg.de)) and synthesized by Integrated DNA Technologies (IDT, Coralville, IA, USA). The guide RNA (gRNA) was mixed with tracrRNA (Alt-R CRISPR-Cas9 tracrRNA, IDT) and Cas9 protein (Alt-R S.p. Cas9 Nuclease V3, IDT) and injected into one-cell-stage medaka embryos. The injected mixture comprised 3 μM gRNA, 3 μM tracrRNA, and 10 μg/L Cas9 protein solutions (ratio 4.5:4.5:1). The microinjection apparatus used was a Nanoject II (Drummond Scientific Co., Broomall, PA, USA) and 1 mm glass capillaries (Narishige, Tokyo, Japan, G-1) formed using a horizontal pipette puller. Injected embryos were cultured at 28˚C for approximately 10 d, after which the hatched larvae were genotyped to establish stable mutant strains using genomic DNA extracts from their membranous fins, as described in a previous study [32].

## Histological analysis

Medaka suspected of being unable to survive till 40–60 weeks of age (3 wildtype and 10 $rad50^{\Delta2/+}$ and 2 $rad50^{\Delta9/+}$ medaka from the $G_0$ and $F_1$ generations) were immediately euthanized using 0.06% tricaine methane-sulfonate [33] and examined histopathologically. Fish used in the rheotaxis analysis were randomly selected (six wildtype and $rad50^{\Delta2/+}$ medaka each) and examined. Fish were fixed in Bouin's fluid (Muto Pure Chemicals Co., Ltd., Tokyo, Japan) and shaken at room temperature (23–26˚C) for 48 h, following which the solution was changed to a 10% formalin neutral buffer solution and shaken again at room temperature for 24 h. Fixed fish were dehydrated in absolute ethanol and separated into head and trunk parts using a surgical knife. Each part was embedded in paraffin. The eyes, thyroid glands, and brains were observed using coronal sections from the head parts. To observe internal organs, sagittal sections were obtained from the trunks and stained using hematoxylin–eosin. Digital images were captured using AS ONE microscope and AS ONE PCM500 digital camera (AS ONE Co. Ltd., Osaka, Japan).

## Survival curve

Wildtype ($n = 7$, control for $G_0$ generation; $n = 15$, control for $F_1$ generation; $n = 17$, control for $F_2$ generation), $rad50^{\Delta2/+}$ ($n = 7$ for G0 generation; $n = 16$ for $F_1$ generation; $n = 33$ for $F_2$ generation), $rad50^{\Delta9/+}$ ($n = 10$ for $F_1$ generation), and $rad50^{\Delta2/\Delta2}$ ($n = 18$ for $F_2$ generation) medaka were reared in each of 4.5 L aquaria with a density of up to 10 fish per aquarium. Feed was provided thrice a day, with sufficient feed for all fish to consume in 15 min without any feed remaining at the bottom of the aquarium. Fish that died within 70 weeks from hatching were recorded. Some $rad50^{\Delta2/+}$ medaka older than 40 weeks showed abnormal swimming with enlargement of the thyroid gland and capillary congestion inside eyes. These fish, which found it difficult to survive and most died within a week, were classified as "dying fish." Samples from fish aged 40–60 weeks were fixed as previously described in histological analysis.

## Rheotaxis analysis

Rheotaxis was performed according to Takeuchi et al. [34]. Wildtype ($n = 12$) and $rad50^{\Delta2/+}$ medaka from $F_3$ generation ($n = 13$) were used for a quantitative comparison of rheotaxis. The intensities of the amplitude of the left and right body movement (body position) and left and right head movement around the center of the body (head yawing) under constant water flow were used as indices. Water flow velocity based on standard body length [SL, the length of a fish measured from the tip of the snout to the posterior end of the last vertebra; no significant difference between $SL_{wildtype}$ (32.7 ± 3.0 mm) and $SL_{mutant}$ (33.3 ± 2.8 mm)), Mann–Whitney U test; $p = 0.6444$), and 2 × and 3 × SL/s were used for this analysis. Twenty-week-old medaka with no pathological findings in the eyes or internal organs were placed in a swim-mill

chamber (10 cm along the y-axis, parallel to the flow direction, and 2 cm along the x-axis), and the flow velocity was set from 2 to 3 × SL/s. Swimming at each velocity was recorded for 10 s using a video recorder (DMK27BUR0135, The Imaging Source, Greenville, SC, USA) located above the chamber (Fig 4A). Positional information of the medaka was used as the output, with the swimming direction as the x-axis and the direction perpendicular to the swimming direction as the y-axis. Next, the change in the y-axis for 10 s was graphed (Fig 4B) as an example, and the change in the y-axis was calculated as a root mean square (RMS) value. Body and head positions were tracked using DeepLabCut ver. 2.0. [35,36], and their coordinates were extracted (S1 Information). The RMS value of the body position was calculated based on the coordinates of the body positions. For head yawing, the RMS value was calculated based on the coordinates of head positions minus those of body positions.

## Measurement of hindbrain size

The size of the hindbrain was quantitatively compared using images of the dorsal side of the brain. After anesthetizing medaka with 0.06% MS-222, the frontal bone was removed using scissors and tweezers to observe the telencephalon, midbrain, and hindbrain. The brains were photographed using a stereomicroscope with a digital camera (Stemi 305, Zeiss, Oberkochen, Germany). Next, the areas of the telencephalon, midbrain, and hindbrain in the image were calculated using ImageJ software (version 1.53e; Bethesda, MD, USA). The area of the hindbrain/(area of the telencephalon + midbrain + hindbrain) was then calculated and used as an index value for a size assessment of each medaka hindbrain.

## Measurement of Purkinje cell number

Six wildtype and *rad50*$^{\Delta2/+}$ medaka each were randomly selected from the medaka used for rheotaxis analysis to prepare hematoxylin–eosin (HE)-stained specimens of the hindbrain. Purkinje cells in a 400 μm square range, including the boundary between the Purkinje cell layer and the granular layer on the specimens, were counted (Fig 5A).

## Statistical analysis

R software (version 4.0.1) was used for statistical analyses. The chi-square test was used for comparative analysis of the genotype of the offspring from *rad50*$^{\Delta2/+}$ crosses and the percentages of Purkinje cells in the granular layers of the hindbrain. Differences in RMS values and hindbrain size between wildtype and *rad50*$^{\Delta2/+}$ medaka were analyzed using the Mann–Whitney U test. Changes in the number of surviving fish were estimated using Kaplan–Meier survival curves, whereas the difference in survival curves was analyzed using the log-rank test together with the Bonferroni correction. Differences between the averages of 50% survival times of wildtype and *rad50*$^{\Delta2/+}$ medaka were analyzed using Welch's t-test. Statistical significance was set at $p < 0.05$.

# Results

## Mutants

The exon composition of the medaka *rad50*, based on a previous study [1], is illustrated in Fig 1A. With the expectation that the insertion of the mutation downstream of the putative start codon would cause a functional defect, the mutation site was set to the coiled-coil region that precedes the hook construct of medaka *rad50*. In detail, gRNA corresponding to bases 21–40 of exon 11 of *rad50* of STIII medaka was designed (Fig 1B: two-way arrow and double underline). Of the 38 embryos injected with a mixed solution of gRNA and Cas9 nuclease, 24

developed and hatched normally. Mutations in medaka *rad50* were detected in 19 of the hatched larvae, and juveniles with 2- or 9-base pair deletion heterozygotes (*rad50^{Δ2/+}* or *rad50^{Δ9/+}*) were selected and raised. The gene with the 2-base pair deletion was estimated to have a stop codon between the bases 85 and 87 of exon 11 (Fig 1B; asterisk of 'Δ2'), which led to a lack of the hook construct required for rad50 dimer formation causing a functional deficiency. The gene with a 9-base pair deletion (Fig 1B; 'Δ9') was expected to express rad50, which would lack three amino acids but still partly retain its function.

## Phenotype of mutants

No difference in growth was observed among the three groups, and no significant difference was observed in the body weights of the dead medaka at 40–60 weeks (326.0 ± 10.5 mg for wildtype [$n = 5$]; 329.1 ± 81.1 mg for *rad50^{Δ2/+}* medaka [$n = 13$]; 312.0 ± 38.4 mg for *rad50^{Δ9/+}* medaka [$n = 4$]; p = 0.9046; one way-ANOVA). Further, 13 out of the 23 *rad50^{Δ2/+}* mutant fish aged 40–60 weeks ($n = 7$ for the $G_0$ generation; $n = 16$ for the $F_1$ generation) were identified as either dead or dying. All dying *rad50^{Δ2/+}* medaka were either lying on the bottom of the water or swimming abnormally. Some had capillary congestion inside the eyes and enlarged thyroid glands. No obvious abnormalities were observed among the wildtype and *rad50^{Δ9/+}* medaka.

## Histological findings

Macro- and microscopic histological findings of *rad50^{Δ2/+}* mutants that had died or were dying at ages 40–60 weeks are shown in Table 1 and described below. Furthermore, one or both eyes of 4 out of the 13 dead or dying fish were swollen (Fig 2A: right eye is shown). A reddish mass was observed at the base of each swollen eye (Fig 2A; arrowhead). Capillary hemangiomas were formed around the choroids of these eyes in 2 out of 10 histologically analyzed fish. Hemangiomas were also observed in the retina (Fig 2C and 2E; arrows #1). Telangiectasias were observed in the hemangiomas and near the pigment epithelial layer (Fig 2C and 2E; arrows #2). The structures of these retinas collapsed (Fig 2C), and many histiocytes were scattered between the rod/pyramidal cell layer and the pigment epithelial layer (Fig 2C and 2E; arrow #3). The thyroid glands of 4 out of the 10 histologically analyzed fish of *rad50^{Δ2/+}* mutants are shown in Fig 2B (arrowhead). Four fish showed proliferation by follicle epithelial cells (Fig 2F; arrow), which formed nodular thyroid hyperplasia (Fig 2H). In addition, some granulomatous nodules (Fig 2I and 2J; arrow #1) and many histiocytes (Fig 2J; arrow #2) were observed in the livers of three fish. Granulomas (Fig 2L and 2M; arrows #1) and hemangiomas (Fig 2E, arrow #2; Fig 2N, magnified view) were observed in the kidney, and telangiectasias were observed in the hemangiomas. A structure dominated by adrenal gland-derived cells was observed in the kidneys of one of the 10 histologically analyzed fish (Fig 2O). Proliferated adrenal-derived cells (Fig 2P; adrenal cortex-derived cells indicated by arrow #1 and adrenal medulla cells indicated by arrow #2) and infiltrated histiocytes (Fig 2P; arrow #3) were observed in this structure. Thus, one or two of the above-mentioned tumorous lesions were observed in 8 out of the 10 histologically analyzed fish of the *rad50^{Δ2/+}* mutants and telangiectasias in 6 of the mutant fish. Three of the wildtypes and two of the *rad50^{Δ9/+}* mutants were observed histologically, but no histological abnormalities were observed (Fig 2D, 2G, 2K and 2Q).

## Survival time

Survival curves corresponding to a period of 70 weeks after hatching of wildtype, *rad50^{Δ2/+}*, and *rad50^{Δ9/+}* medaka in the $F_1$ generation were compared. A significant difference was observed among the three curves (Fig 3A; $n = 15$ for wildtype; $n = 16$ for *rad50^{Δ2/+}*; $n = 10$ for *rad50^{Δ9/+}*; $p = 0.002$, log-rank test with Bonferroni correction), as well as between the median

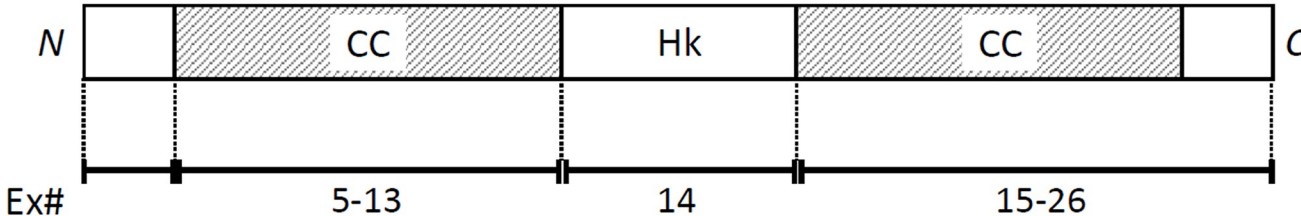

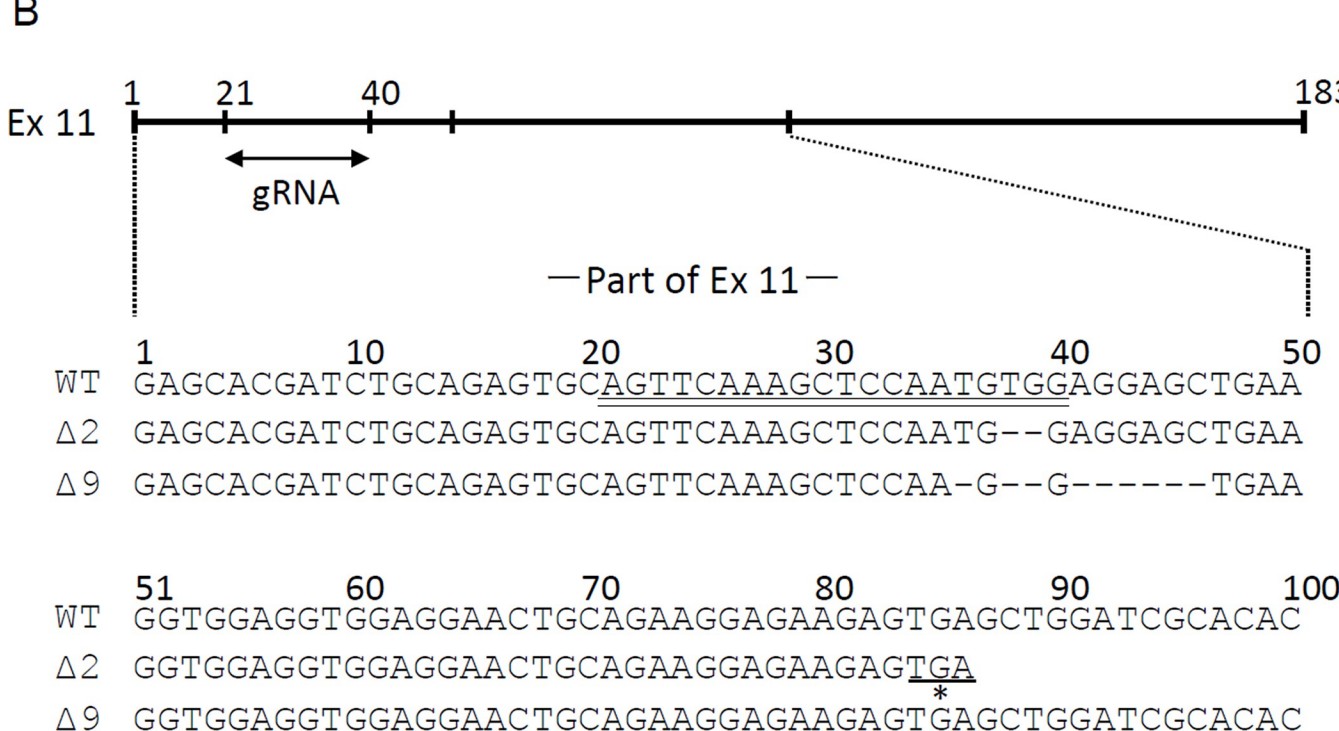

**Fig 1. Location of guide RNA (gRNA) design and nucleotide sequence generation using the CRISPR/Cas9 system. (A)** Predicted domain structure of medaka rad50 protein and the corresponding exon order. 'CC' and 'Hk' indicate the coiled-coil region and the hook construct, respectively. '*N*' and '*C*' indicate the N- and C- termini of the rad50 protein, respectively. The exon number (Ex#) corresponding to each area is displayed at the lower position of the predicted domain structure. **(B)** Location of gRNA design and multiple sequence alignments of wildtype (WT) and mutant sequences. "Ex11" indicates "exon 11." The 183-base sequences of Ex11 are schematically displayed to the right. A gRNA that recognizes the 21st to 40th bases was designed (two-way arrow). Parts of the base sequences of Ex11 of WT, 2-base pair deletion, and 9-base pair deletion were aligned (shown at the lower end of the figure). The sequence recognized by gRNA is double-underlined in the WT sequence. Stop codons in the sequence are underlined or marked with asterisks.

survival times for wildtype, $rad50^{\Delta9/+}$, and $rad50^{\Delta2/+}$ medaka (66.5, 65.5, and 56.0 weeks, respectively). The survival curve of $rad50^{\Delta9/+}$ medaka was similar to that of wildtype. Additionally, in the other generations ($G_0$ and $F_2$), the number of dead or dying $rad50^{\Delta2/+}$ fish increased at around 40 weeks of age, and significant differences were observed between the survival times of wildtype and $rad50^{\Delta2/+}$ groups ($p < 0.01$; log-rank test). The average median survival times of $G_0$, $F_1$, and $F_2$ generations of both wildtype and $rad50^{\Delta2/+}$ medaka were 65.7 ± 1.1 and

**Table 1. Macroscopic and microscopic findings of dead *rad50*$^{\Delta2/+}$ mutant fish (40-60 weeks of age).**

| Fish | Macroscopic pathological findings | | Microscopic pathological findings | | | | | | | | |
| | Eye enlargement | Thyroid nodule | Retina | | | Liver | Kidney | | | | Thyroid |
| | | | Capillary hemangioma | Telangiectasia | Histiocyte infiltration | Granuloma and histiocyte infiltration | Capillary hemangioma | Telangiectasia | Granuloma and histiocyte infiltration | Adrenal cell proliferation | Nodular thyroid hyperplasia |
|---|---|---|---|---|---|---|---|---|---|---|---|
| A | - | - | - | - | - | - | - | - | - | - | - |
| B* | - | - | --- | --- | --- | --- | --- | --- | --- | --- | --- |
| C | - | + | - | - | - | - | - | - | - | - | + |
| D* | - | - | --- | --- | --- | --- | --- | --- | --- | --- | --- |
| E | - | - | - | - | - | + | + | + | + | + | - |
| F | + | - | + | + | + | + | - | - | - | - | - |
| G | + | - | + | + | + | - | - | - | - | - | - |
| H | + | + | - | + | - | - | - | - | - | - | + |
| I | + | - | - | + | - | - | - | - | - | - | - |
| J | - | - | - | - | - | - | - | - | - | - | - |
| K | - | + | - | - | - | + | - | - | - | - | + |
| L | - | + | - | - | - | - | + | + | + | + | + |
| M* | - | - | --- | --- | --- | --- | --- | --- | --- | --- | --- |
| Number of positives | 4 | 4 | 2 | 4 | 2 | 3 | 2 | 2 | 2 | 2 | 4 |

*These three fish could not be analyzed microscopically because they had advanced postmortem degeneration prior to fixing the tissues with Bouin's fluid.

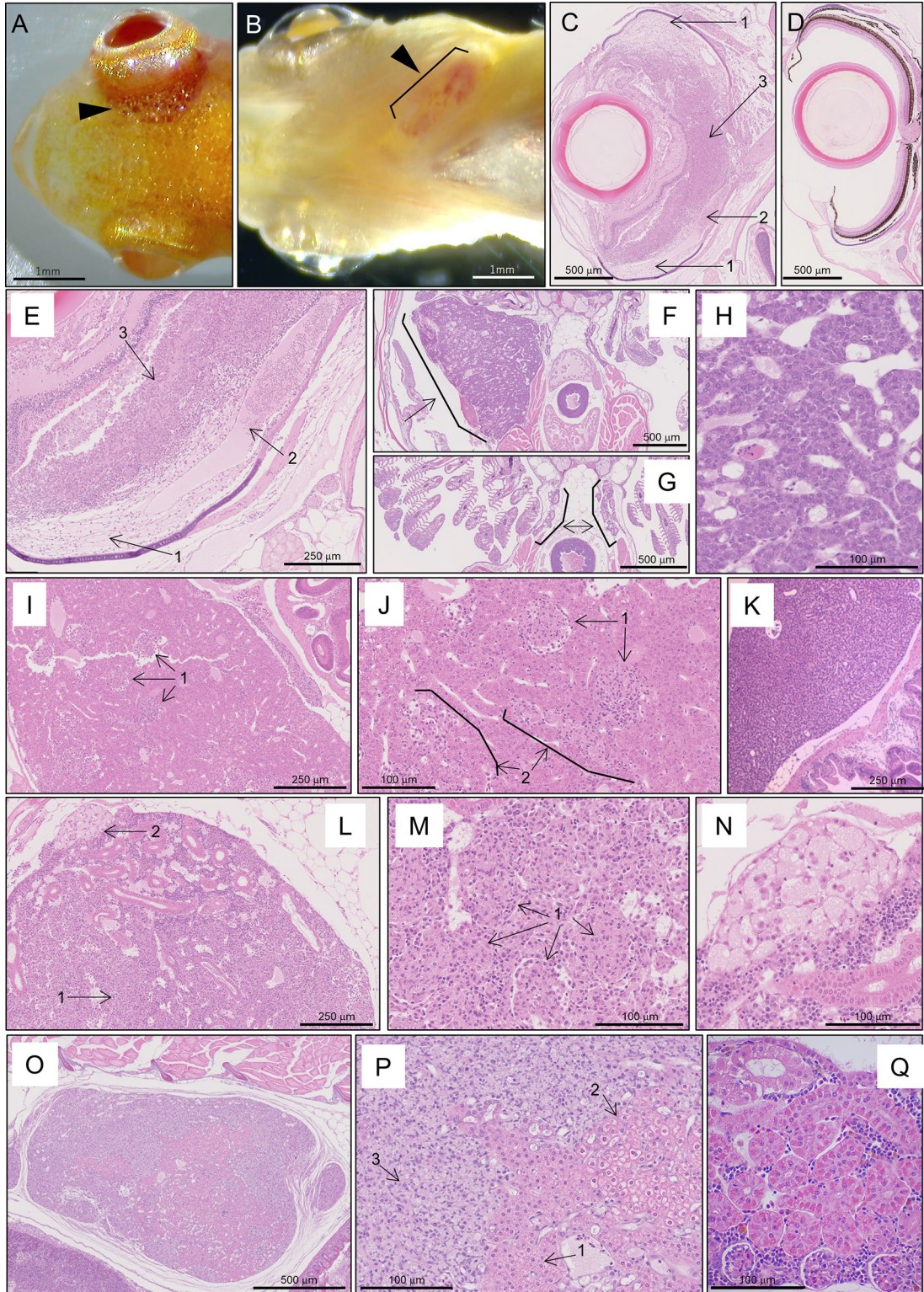

**Fig 2. Macroscopic and microscopic observations of *rad50*$^{\Delta2/+}$ mutants.** (**A**) and (**B**) Macroscopic images of *rad50*$^{\Delta2/+}$ mutants. (**A**) Dorsal view of the head of the mutant and congestion inside the eye (black arrowhead). (**B**) Ventral view of the head of the mutant and enlarged thyroid gland (black arrowhead). (**C–Q**) Microscopic images stained with hematoxylin–eosin (HE). (**C**), (**E**), (**F**), (**H–J**), and (**I–P**) images of *rad50*$^{\Delta2/+}$ mutants. (**D**), (**G**), (**K**), and (**Q**) images of wildtype medaka. (**C**) Right eye with collapsed retina; capillary hemangiomas (arrows #1). (**D**) Right eye with normal retina. (**E**) Magnified view of the lower area of

panel (**C**). Capillary hemangioma (arrow #1), telangiectasia (arrow #2), histiocytes (arrow #3). (**F**) Enlarged thyroid gland (an arrow). (**G**) Normal thyroid gland (two-way-arrow) from wildtype medaka. (**H**) Magnified view of the lower right area of the enlarged thyroid gland in panel (**F**), consisting of nodules formed by thyroid follicular cells. (**I**) Liver with granulomatous nodules; granulomatous constituents (arrows #1). (**J**) Magnified view of the center area of panel (**I**). Granulomatous constituents (arrows #1) and infiltrated histiocytes (arrows #2). (**K**) Normal liver from wildtype medaka. (**L**) Kidney with extensively infiltrated histiocytes (arrow #1) and hemangioma (arrow #2). (**M**) Magnified view of the panel (**L**). Granulomatous constituents (arrows #1). (**N**) Magnified view of the area shown by arrow #2 in panel (**L**). (**O**) Structure dominated by adrenal gland-derived cells. (**P**) Magnified view of panel (**O**). Adrenal cortex-derived cells (arrow #1), adrenal medulla cells (arrow #2), and infiltrated histiocytes (arrow #3). The histological image provided by histological analysis of the other wildtype medaka (two fish) and the *rad50*$^{\Delta 9/+}$ mutants (two fish) was not abnormal, similar to those shown in panels (**D**), (**G**), (**K**), and (**Q**).

54.2 ± 2.6 weeks, respectively (Welch's *t*-test; $p = 0.001$). The ratio of the number of larvae from in-crosses of *rad50*$^{\Delta 2/+}$ medaka was wildtype:*rad50*$^{\Delta 2/+}$:*rad50*$^{\Delta 2/\Delta 2}$ = 22:46:18, respectively, which was not significantly different from the Mendelian ratio ($p = 0.734$; Chi-square test). However, the homozygous mutants hardly survived after 4 weeks of hatching (Fig 3B), and their 50% survival time was 6 weeks. The swimming was unstable even in rearing water without water flow. The reduced lifespan of *rad50*$^{\Delta 2/\Delta 2}$ homozygotes made it difficult to maintain the samples before analyses. *rad50*$^{\Delta 9/+}$ medaka were subjected to survival time measurement and histological analyses to determine whether genome-editing frameshift mutations without putative premature stop codons affected lifespans and tumorigenesis. However, these homozygotes could not be analyzed in detail because of their smaller sample sizes. In-crosses of *rad50*$^{\Delta 9/+}$ medaka produced only three *rad50*$^{\Delta 9/\Delta 9}$ medaka with 63, 68, and 70 weeks or more lifespans, which were close to those of wildtype medaka, suggesting that the 9-base pair deletion did not affect the lifespan of medaka.

## Rheotaxis analysis

Rheotaxis abilities were quantitatively compared to investigate the effect of the 2-base pair deletion of *rad50* on medaka hindbrains. This was based on the fact that fish with normal hindbrains are able to maintain their body position while swimming under constant water flow conditions. At a flow velocity of 2 × SL/s, the RMS value of *rad50*$^{\Delta 2/+}$ medaka was significantly higher than that of wildtype medaka (Fig 4C; Mann–Whitney U test; $p = 0.036$). However, with increased flow velocity (3 × SL/s), the difference between the RMS values of *rad50*$^{\Delta 2/+}$ and wildtype medaka became greater (Mann–Whitney U test; $p < 0.001$), indicating that *rad50*$^{\Delta 2/+}$ medaka were inferior to wildtype medaka in terms of rheotaxis ability. With respect to head yawing at a flow velocity of 2 × SL/s, no significant difference was observed between the RMS values of *rad50*$^{\Delta 2/+}$ and wildtype medaka (Fig 4D; Mann–Whitney U test; $p = 0.8065$). However, at a flow velocity of 3 × SL/s, the RMS value of *rad50*$^{\Delta 2/+}$ medaka was significantly higher than that of wildtype medaka (Mann–Whitney U test; $p < 0.001$). S1 and S2 Movies show representative wildtype and *rad50*$^{\Delta 2/+}$ medaka, respectively, swimming at a velocity of 3 × SL/s. There was no significant difference between *rad50*$^{\Delta 2/+}$ and wildtype medaka at a flow velocity of 1 × SL/s. The medaka could not swim for 10 s under a flow velocity of 4 × SL/s.

## Hindbrain

No significant difference was observed between the hindbrain sizes of wildtype and *rad50*$^{\Delta 2/+}$ medaka (S1 Fig). Starting from the outside of wildtype medaka hindbrains, molecular, Purkinje cell, and granular layers were arranged in that order (Fig 5B). In *rad50*$^{\Delta 2/+}$ medaka, granular cells were dispersed in the molecular layer, whereas Purkinje cells were dispersed in the granular layer (Fig 5C; black and white arrowheads). On average, 6.4% ± 3.7% and 32.1% ±

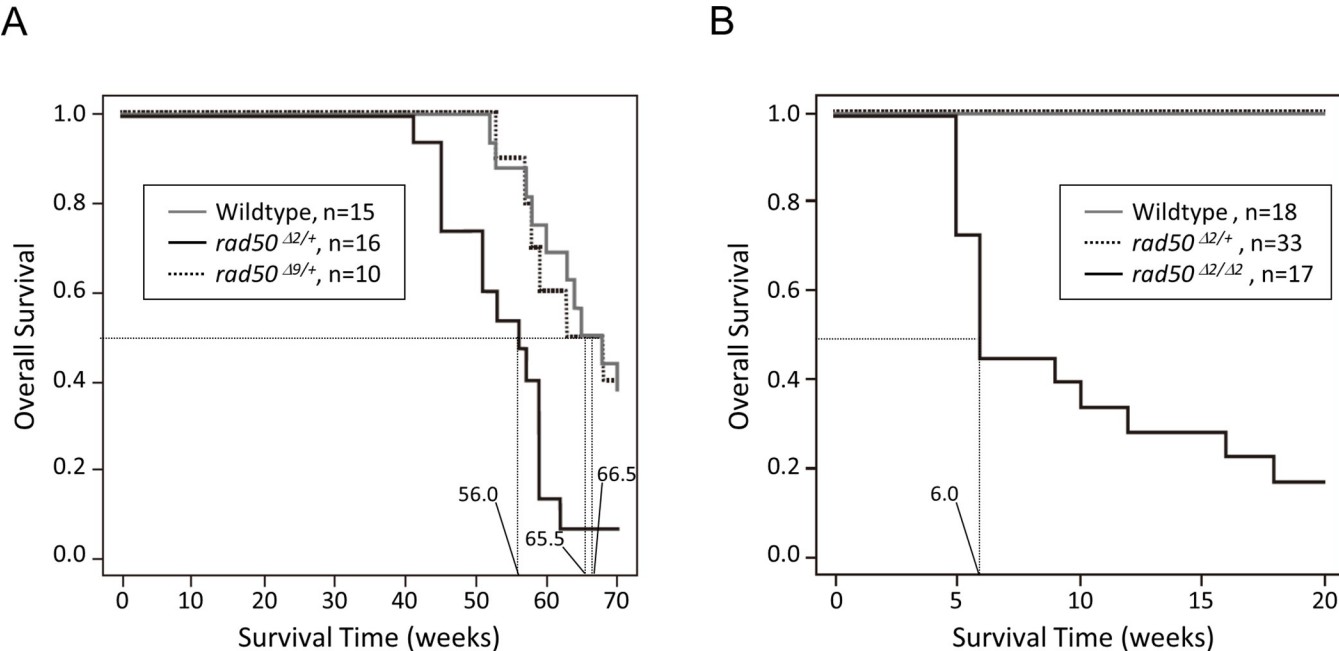

**Fig 3. Kaplan–Meier survival curves of heterozygous and homozygous mutants.** (**A**) The curves of wildtype, *rad50*$^{\Delta2/+}$, and *rad50*$^{\Delta9/+}$ medaka of F$_1$ generation (*n* = 15, 16, and 10, respectively). (**B**) The curves of wildtype, *rad50*$^{\Delta2/+}$, and *rad50*$^{\Delta2/\Delta2}$ medaka of F$_2$ generation (*n* = 18, 33, and 17, respectively). Post-hatching time is shown in weeks, and survival numbers recorded up to 70 weeks after hatching (**A**) or 20 weeks after hatching (**B**). Observed differences in survival curves among wildtype, *rad50*$^{\Delta2/+}$, and *rad50*$^{\Delta9/+}$ medaka of panel (**A**) and in survival curves among wildtype, *rad50*$^{\Delta2/+}$, and *rad50*$^{\Delta2/\Delta2}$ medaka of panel (**B**) were significant (*p* < 0.01 and *p* < 0.001, respectively; log-rank tests with Bonferroni's correction).

20.5% of Purkinje cells were dispersed in the granular layer of wildtype and *rad50*$^{\Delta2/+}$ medaka, respectively; the difference was significant (*p* < 0.001; Chi-square test; Table 2). Furthermore, spindle-shaped (tear-shaped) Purkinje cells were observed in *rad50*$^{\Delta2/+}$ medaka (Fig 5C, black arrowheads; Fig 5D; arrows).

**Table 2. Numbers and percentages of Purkinje cells in granular/Purkinje and monocular layers.**

| Fish | | In granular layer | | In Purkinje & molecular layers | |
|---|---|---|---|---|---|
| | | **N** | **(%)** | **N** | **(%)** |
| Wildtype | N | 6 | 5.7 | 100 | 94.3 |
| | O | 5 | 6.3 | 74 | 93.4 |
| | P | 3 | 2.5 | 116 | 97.5 |
| | Q | 3 | 3.0 | 96 | 97.0 |
| | R | 4 | .2 | 45 | 91.8 |
| | S | 7 | 12.7 | 48 | 87.2 |
| Average | | 4.7 ± 1.6 | 6.4 ± 3.7* | 79.8 ± 29.1 | 93.5 ± 3.8* |
| *rad50*$^{\Delta2/+}$ | T | 44 | 58.7 | 31 | 41.9 |
| | U | 27 | 49.1 | 28 | 50.9 |
| | V | 12 | 41.4 | 17 | 58.6 |
| | W | 12 | 20.0 | 48 | 80.0 |
| | X | 13 | 15.1 | 73 | 84.9 |
| | Y | 7 | 8.0 | 80 | 92.0 |
| Average | | 19.2 ± 13.9 | 32.1 ± 20.5* | 46.2 ± 25.6 | 68.1 ± 20.3* |

*\*p* < 0.001, Chi-square test using mean percentages.

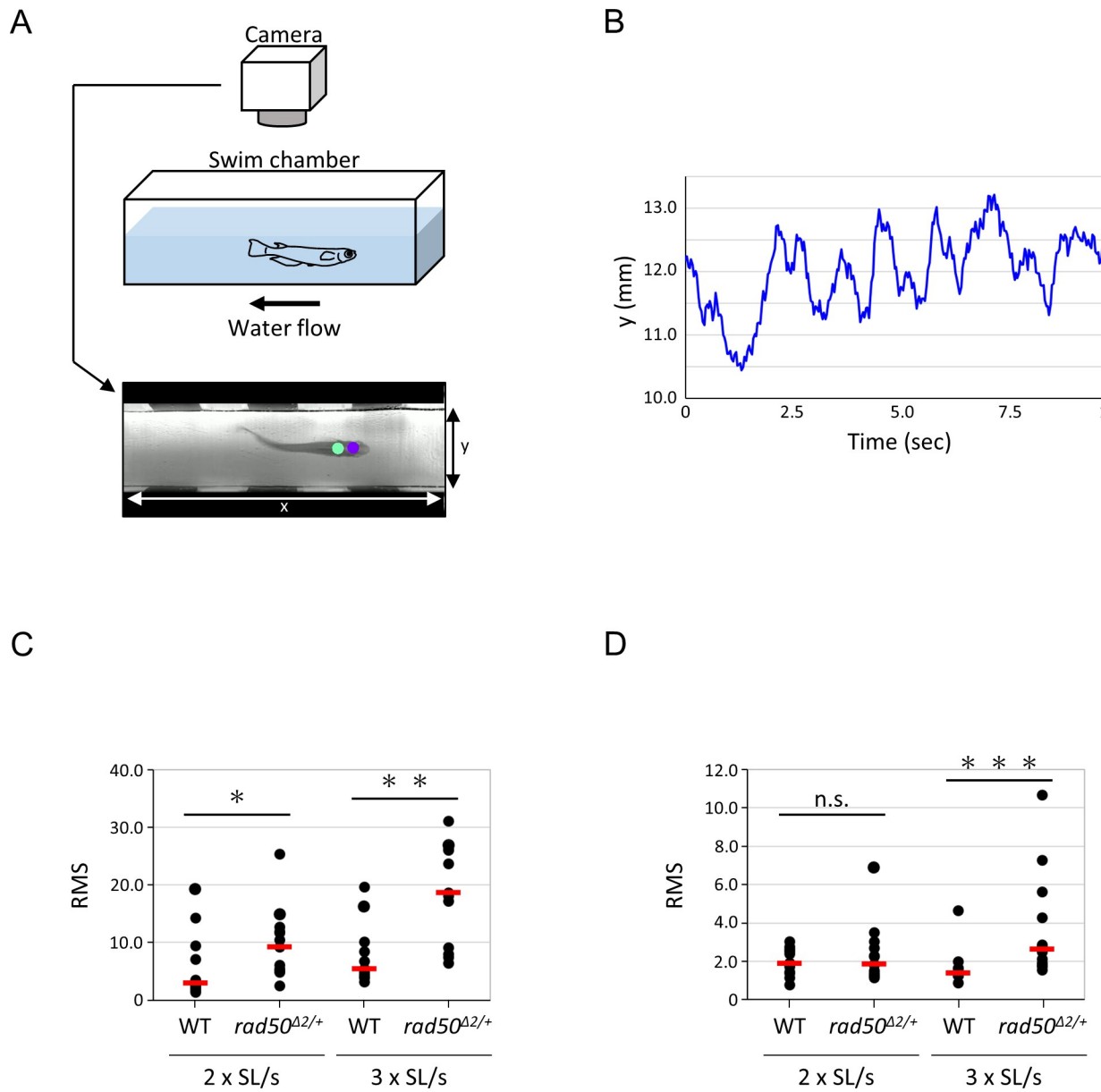

**Fig 4. Rheotaxis analysis of *rad50*ᴬ²/⁺ medaka.** (**A**) Experimental setup and a representative photographic image of swimming behavior. The purple and green circles in the image represent tracking of the head and body positions. (**B**) Representative data from a wildtype medaka showing a change in body position (y-axis) for 10 s. (**C**) or (**D**) A graph showing the root mean square (RMS) of body position (**C**) and head yawing (**D**) for wildtype (*n* = 12) and *rad50*ᴬ²/⁺ medaka (*n* = 13) in water flow velocities of 2 and 3 × standard body length (SL)/s. The black dots and short horizontal red lines represent the data obtained from one individual and the median, respectively. *, **, ***, and n.s. indicate $p < 0.05$, $p < 0.01$, $p < 0.001$, and the absence of a statistically significant difference, respectively (Mann–Whitney U test).

## Discussion

To the best of our knowledge, this is the first study to successfully generate a model fish carrying mutant *rad50*. To ensure a null mutation of medaka *rad50*, we used genome editing to target the position of the coiled-coil region preceding the hook construct to generate a stop codon close to the back of that position. With the loss of the hook construct, *rad50* stopped expressing functional RAD50 [11,37–39]. Therefore, the 2-base pair deletion in *rad50* of

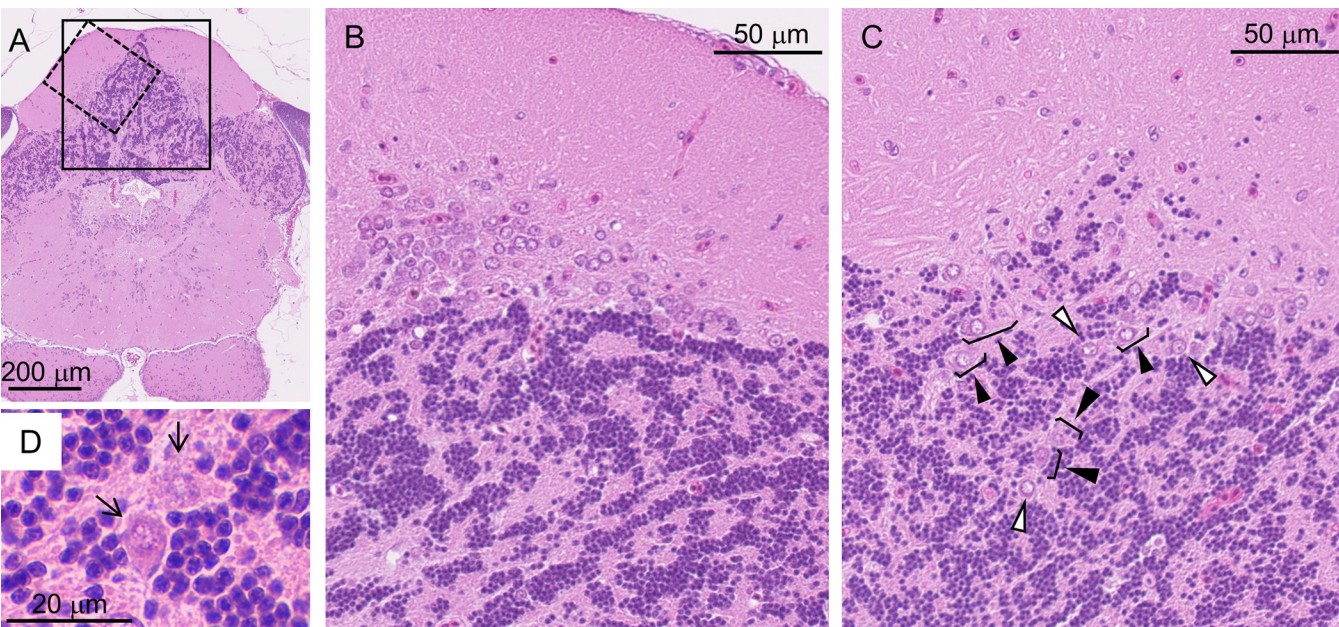

**Fig 5. Microscopic observation of *rad50*<sup>Δ2/+</sup> medaka hindbrain.** (**A**) Photographic image showing the area in which the number of Purkinje cells was measured (black lined square, 400 μm × 400 μm) and the positions of panels (**B**) and (**C**) (dashed square). (**B**) A representative image of wildtype medaka. (**C**) A representative image of *rad50*<sup>Δ2/+</sup> mutants. Purkinje cells dispersed in the granular layer (black and white arrowheads). Spindle-shaped (tear-shaped) Purkinje cells (black arrowheads). (**D**) A magnified view of spindle-shaped (tear-shaped) Purkinje cells (arrows) from panel (**C**).

*rad50*$^{Δ2/+}$ medaka would have resulted in the loss of functional rad50. In contrast, a 9-base pair deletion in *rad50* that caused a small in-frame deletion of rad50 retained some rad50 function in *rad50*$^{Δ9/+}$ medaka. Furthermore, STIII medaka is a transparent fish in which morphological abnormalities, including organ defects and tumorigenesis, can be easily observed. Therefore, *rad50*$^{Δ2/+}$ and *rad50*$^{Δ9/+}$ STIII medaka were deemed to be appropriate models that could be used to evaluate the phenotypes affected by *rad50* mutations. All *rad50*$^{Δ2/Δ2}$ medaka were unstable in swimming even under normal conditions without water flow, suggesting the possibility that the rheotaxis ability of the homozygous mutants was lower than that of heterozygotes. On the contrary, no abnormalities in swimming and lifespan were observed in *rad50*$^{Δ9/Δ9}$ medaka in a normal rearing condition, although those with a small sample size were preliminarily analyzed, suggesting that the 2 bp deletion is more harmful than the 9 bp deletion in terms of swimming abilities and lifespan of medaka. Given the short survival time and small sample sizes of the homozygotes, the effects of *rad50* frameshift mutation with premature translation stop on tumorigenesis, lifespan, A-T, etc., were mainly discussed based on the comparison between wildtype and *rad50*$^{Δ2/+}$ medaka.

Capillary hemangiomas of the retina, kidney, and nodular thyroid hyperplasia were observed in *rad50*$^{Δ2/+}$ medaka. These results may substantiate the reported tumorigenic properties of RAD50, as suggested by the prevalence of *RAD50* mutations in patients with cancer. A study on Chinese patients with breast cancer demonstrated no association between *RAD50* mutation and the morbidity associated with the disease [40]. In contrast, a study in Finnish patients with breast and ovarian cancer reported a significant association between *RAD50* 687delT and predisposition to cancer in Finnish people [41]. Concordant with this previous study [41], we identified a *RAD50* I505fs germline nonsense mutation in the *RAD50* gene, indicating an association of *RAD50* mutation with the cancers in the 37-year-old patient presented with concomitant duodenal and rectal cancers. The exon containing the mutation in

the patient corresponds to the position where the stop codon was induced using the CRISPR/Cas9 system in the current study. Therefore, our medaka mutant demonstrated that, as shown by experimental animals with other defective MRN-ATM pathway molecules, the loss-of-function mutation of *rad50* would have tumorigenic properties; however, further studies are required to elucidate the clinical significance of the *RAD50* mutation in human cancers.

Epidemiological studies have indicated that mutations in DNA repair genes are involved in lifespan and aging [42]. However, evidence regarding the effect of *RAD50* mutations on lifespan is inadequate. Mouse early embryonic stem cells homozygous for mutated alleles of *rad50* are nonviable [10]; the study showed no distinctive features regarding growth rate, viability, and fertility in mice heterozygous for the mutant allele compared to wildtype. Bender et al. (2002) demonstrated a decrease in growth rate but did not show the lifespan data of the mouse hypomorphic RAD50 model [43]. The results of the current study indicated that the survival time of heterozygous mutants was shorter than that of wildtype medaka, even in environments free of exposure to mutagens, such as UV light and carcinogens. The short lifespan could be attributed to the *rad50* mutation and/or mutation-associated histological changes occurring after approximately 40 weeks of age after hatching. The hatched larvae from a cross between heterozygotes exhibited a Mendelian genotypic ratio. As opposed to zebrafish studies that demonstrated that embryonic development was affected by *rad50* [28], our results suggest that homozygous mutations in *rad50* may not affect embryogenesis and hatching in a stable environment. However, normal rad50 appears essential for survival in the early stages of development following hatching because the survival time in homozygous mutants was considerably shorter after hatching than before.

*RAD50* mutations were detected in two patients with partial symptoms of NBS, classified as NBSLD [8,9]. Mutations in *RAD50* will arguably lead to some of the same symptoms of A-T, as do ATLD1 and NBS, having been caused by mutations in other components of the MRN/ATM pathway. Histological analysis in *rad50*$^{\Delta 2/+}$ medaka showed an association between *rad50* and telangiectasia. Telangiectasia, which are found on the surface of the skin or the edges of eyeballs [13], have been observed in patients with A-T [12]; however, telangiectasia have not been previously observed in experimental animals with mutations in *atm*, *mre11a*, or *nbs1* genes [11,43]. To the best of our knowledge, *rad50*$^{\Delta 2/+}$ medaka is the first experimental animal model exhibiting telangiectasia, similar to that found in patients with A-T. In this study, the rheotaxis ability against flow velocity was used as an evaluation index for ataxia. The hindbrain of fish, which corresponds to the cerebellum of mammals, integrates processes such as vestibular motor reflexes, gaze retention, and lateral line signal detection to maintain posture against flow velocity [44–46]. Several experiments using zebrafish and medaka have demonstrated that deficiencies in the genes required for hindbrain homeostasis may reduce rheotaxis ability [34]. Mice with disordered Purkinje and granular cells in the cerebellum showed poor performance in the rotarod test, which measures coordination and motor learning [47]. Therefore, disordered cells in the *rad50*$^{\Delta 2/+}$ medaka hindbrain may lead to deterioration of rheotaxis ability. In addition, spindle-shaped Purkinje cells that were not observed in the wildtype were characteristically observed in *rad50*$^{\Delta 2/+}$ medaka. The proximal axons of Purkinje cells in the cerebellum swell into spindle-shaped Purkinje cells (torpedo) in patients with A-T [48], suggesting that the torpedo in *rad50*$^{\Delta 2/+}$ medaka may also be associated with defects in their rheotaxis ability. These results demonstrated that in vertebrates, associations exist between the *rad50* mutation and ataxia or telangiectasia.

Histological analysis in our mutant supported data associated with immunosuppression in mice with hypomorphic RAD50 [11]. Although no immune impairment has been observed in the two patients with NBSLD, some patients with A-T show immune disorders, and mutant mice with hypomorphic RAD50 showed defects in primitive hematopoietic cells [11,43].

Histiocyte and lymphocyte infiltration in $rad50^{\Delta2/+}$ medaka was the abnormal proliferation of cells involved in the lymphatic system. A-T is strongly associated with the abnormal proliferation of cells involved in the lymphatic system, as proliferation weakens the immune system [49]. Our study demonstrated that $rad50^{\Delta2/+}$ medaka showed immunosuppression, which constitutes a part of the findings in patients with A-T. No histological changes were observed in reproductive tissues, and no abnormal mating behavior was observed in $rad50^{\Delta2/+}$ medaka, and the embryogenesis of offspring was normal.

Our study has a few limitations. First, the size of the analyzed cohorts was relatively small. Therefore, further studies with larger sample sizes are warranted. Second, although A-T, ATLD1, NBS, and NBSLD are typically autosomal recessive disorders associated with biallelic mutations of a relevant gene, most results of the present study are based on an analysis of heterozygotes of *rad50* mutation. However, a subset of A-T patients carries a heterozygous *ATM* mutation, suggesting that some type of heterozygous *ATM* deficiency may give rise to the A-T phenotype. Similarly, *RAD50* mutants may exert haploinsufficiency or dominant-negative effects on A-T phenotype development. A previous study also implicated the dose-dependent dominance of a heterozygous *rad50* mutant in a mouse model [10]. In our medaka model, the homozygous mutant of *rad50* was semi-lethal and, thus, inappropriate for the precise analysis of several data sets. Regardless of the zygosity of the mutation, our medaka model exhibiting the A-T phenotype will be useful in future experiments investigating the pathogenesis of the disease.

In conclusion, the analysis of medaka fish revealed that a heterozygous mutation of *rad50* could cause tumorigenesis. Furthermore, the mutation results in a short lifespan, which is one of the signs of A-T, ATLD1, NBS, and NBSLD patients. In addition, to the best of our knowledge, this study is the first to demonstrate that most A-T phenotypes may be concurrently reproduced via a vertebrate *rad50* germline mutation model. The $rad50^{\Delta2/+}$ STIII medaka developed ataxia represented by rheotaxis disability, Purkinje cell disorder in the hindbrain, telangiectasia, immunological abnormalities, and tumor formation, all of which are typical characteristics of A-T patients. Thus, $rad50^{\Delta2/+}$ medaka could contribute to a deeper understanding of the molecular mechanisms underlying *RAD50* mutation-induced tumors and related A-T phenotype, as well as help develop novel therapeutic strategies based on improved insight into molecular disorders.

## Supporting information

**S1 Fig. Hindbrain size.** A ratio of the area of the hindbrain to area of the telencephalon, midbrain, and hindbrain in wildtype (WT) and $rad50^{\Delta2/+}$ medaka. The areas were calculated using images. The black dots and short horizontal red lines represent the data obtained from one individual and the median, respectively. n.s. indicates the absence of a statistically significant difference (Mann–Whitney U test).
(TIF)

**S1 Table. PCR and sequencing primers.**
(DOCX)

**S1 Information. Creation of an analytical model and extraction of the head and body coordinates of swimming medaka from movies via DeepLabCut.** DeepLabCut (version 2.0) was used to create an analytical model of medaka swimming and to extract the coordinates of the head and body positions. First, 19 frames were automatically extracted using the "kmeans" algorithm from the video of one wildtype medaka. In these frames, the head position was labeled between the left and right eyes. The body position was labeled between the left and

right pectoral fins. The size of each marker was 5 μm. Nineteen frames were used as the training dataset in DeepLabCut. Next, DeepLabCut was trained 50,000 times to recognize the position of the head and body of the medaka in the movie based on the training dataset, and an analysis model of medaka swimming was generated. The analytical model of medaka swimming was used to analyze all movies of wildtype and $rad50^{\Delta2/+}$ medaka, and the head and body coordinates were extracted. The root mean square values were calculated from the coordinates using Microsoft Excel. The tail markers shown in the movies were not used in this study. (DOCX)

**S1 Movie. Stable swimming of wildtype.**
(MPEG)

**S2 Movie. Unstable swimming of mutant.**
(MPEG)

**S1 Dataset.**
(XLSM)

**S2 Dataset.**
(CSV)

**S3 Dataset.**
(CSV)

**S4 Dataset.**
(XLSX)

**S5 Dataset.**
(XLSM)

## Acknowledgments

We thank Ms. Naomi Matsunaga (Kyorin University) for her assistance in rearing the medaka; Dr. Tomohisa Katada (Kyorin University) for giving advice for using Nanoject II; Sapporo General Pathology Laboratory Co., Ltd. (Sapporo, Japan) for technical assistance with histopathological analyses; Dr. Satoshi Ansai (Tohoku University) for advice regarding genome editing techniques; and Dr. Masayuki Yoshida (Graduate School of Biosphere Sciences, Hiroshima University) for advice regarding rheotaxis analyses.

## Author Contributions

**Conceptualization:** Kouki Ohtsuka.

**Data curation:** Shinichi Chisada, Kouki Ohtsuka.

**Formal analysis:** Shinichi Chisada, Kouki Ohtsuka.

**Funding acquisition:** Kouki Ohtsuka.

**Investigation:** Shinichi Chisada, Kouki Ohtsuka, Masachika Fujiwara, Satsuki Matsushima.

**Methodology:** Shinichi Chisada, Kouki Ohtsuka, Hiroaki Ohnishi.

**Validation:** Shinichi Chisada, Kouki Ohtsuka.

**Visualization:** Shinichi Chisada, Kouki Ohtsuka.

**Writing – original draft:** Shinichi Chisada, Kouki Ohtsuka, Masachika Fujiwara, Kanae Karita, Hiroaki Ohnishi.

**Writing – review & editing:** Masao Yoshida, Takashi Watanabe.

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
