## [Decision Letter · Decision Letter 0]

27 Jan 2023

PONE-D-22-32016A rad50 germline mutation induces tumorigenesis and ataxia-telangiectasia phenotype in a transparent medaka modelPLOS ONE

Dear Dr. Ohtsuka,

Thank you for submitting your manuscript to PLOS ONE. After careful consideration, we feel that it has merit but does not fully meet PLOS ONE’s publication criteria as it currently stands. Therefore, we invite you to submit a revised version of the manuscript that addresses the points raised during the review process.

We look forward to receiving your revised manuscript.

Kind regards,

Sebastian D. Fugmann, Ph.D.

Academic Editor

PLOS ONE

Journal Requirements:

3. Please note that supplementary tables (should remain/ be uploaded) as separate "supporting information" files"

Additional Editor Comments:

In addition to one external expert in the field, Reviewer #1, I have also carefully assessed the manuscript and there are two main issues that need to be addressed: 1) It is unclear what (if any) the phenotypes of the heterozygous D9/+ and the homozygous D9/D9 mutants are.  While a sentence in the discussion states that "a 9-base pair deletion in *rad50 *that caused a small in-frame deletion of rad50 retained some rad50 function in *rad50*D9*/+ *medaka." there is no data that actually shows any defect in the D9/+ fish.  Furthermore, no data at all is presented on the homozygous D9/D9 fish, and this should either be shown or clearly described why this data is not presented.   2) As the D2/+ clearly shows a phenotype it would be interesting to see what the phenotypes of the homozygous D2/D2 fish are (despite shortened lifespan).  Reviewer #1 thinks that brain phenotype would be particularly important to show.   3) All minor concerns raised by Reviwer # should be addressed by making appropriate changes to the text.  

Reviewers' comments:

Reviewer's Responses to Questions

**Comments to the Author**

1. Is the manuscript technically sound, and do the data support the conclusions?

Reviewer #1: No

2. Has the statistical analysis been performed appropriately and rigorously? 

Reviewer #1: Yes

3. Have the authors made all data underlying the findings in their manuscript fully available?

Reviewer #1: Yes

4. Is the manuscript presented in an intelligible fashion and written in standard English?

Reviewer #1: Yes

5. Review Comments to the Author

Reviewer #1: This is an interesting report on the phenotype of a rad50 frameshift variant in medaka. The authors show that homozygotes have a sharply reduced viability and identify some phenotypic differences in heterozygotes. However, the manuscript remains unclear at severel parts and the description could be improved to show its full value for the scientific community.

Major comment:

The authors generated two mutants, the targeted frameshift deletion variant D2 and a non-targeted in-frame deletion variant D9 (the latter one was also followed as potentially mild protein variant). While homozygotes for the D2 variant had a mean survival of only six weeks, it is not clear what the fate of D9 homozygotes was. But this would be important to judge about the role of D9 in rad50 function. Also, D9 mutants seem to have not been assessed in all assays (e.g. Table 2). Furthermore, D2 homozygotes were apparently not assessed for brain phenotypes. The manuscript might benefit from these additional informations and from a side-by-side comparison of results for heterozygotes and homozygotes for both mutants.

Minor comments:

1. Abstract, line 23: The authors postulate that their model reproduced tumorigenesis. But this seems to be overstated in regard that the heterozygous medaka had no malignant tumours (which is stated in line 408). Therefore, the conclusion in the abstract would need to be reworded.

2. Introduction, line 50: Note that the embryonic lethality in rad50 null mice is not different from the described NBSLD patients as these patients were carrying hypomorphic mutations.

3. Introduction, line 78: It should be clarified whether the identified mutation is a nonsense mutation or a frameshift mutation and whether it occurred in the heterozygous state.

4. Introduction, line 80: Microsatellite instability is not an expected outcome of RAD50 heterozygosity. Has this colon cancer been assessed for mismatch repair deficiency?

5. Material and Methods, line 137: The 2 bp deletion needs to be defined by chromosome position and deleted nucleotides.

6. Discussion, line 414: Note that Heikkinen et al. (Ref. 41) did not report on polymorphisms but on RAD50 truncating variants including a frameshift mutation (RAD50 687delT) that would be similar to the one in this report.

6. PLOS authors have the option to publish the peer review history of their article (what does this mean?). If published, this will include your full peer review and any attached files.

Reviewer #1: **Yes: **Thilo Dörk

---

## [Author Response · Author response to Decision Letter 0]

9 Feb 2023

RESPONSE TO THE EDITOR AND REVIEWER

Dear Editor,

Thank you for evaluating our manuscript. We appreciate the guidance and thoughtful suggestions of the Additional Editor and Reviewer, which helped strengthen the quality of our manuscript. We have read the comments provided by the Additional Editor and Reviewer and revised the manuscript accordingly. Please find our point-by-point responses to all comments below; revisions in the manuscript are highlighted in red.

Editor

Comment 1: It is unclear what (if any) the phenotypes of the heterozygous D9/+ and the homozygous D9/D9 mutants are. While a sentence in the discussion states that "a 9-base pair deletion in rad50 that caused a small in-frame deletion of rad50 retained some rad50 function in rad50D9/+ medaka." there is no data that actually shows any defect in the D9/+ fish. Furthermore, no data at all is presented on the homozygous D9/D9 fish, and this should either be shown or clearly described why this data is not presented. 

Response: Thank you for highlighting this. We agree that a comparison of the results of D9/+ could have strengthened our study to unravel whether genome-editing frameshift mutations without putative premature stop codons affected lifespans and tumorigenesis. Unfortunately, we could not perform side-by-side comparisons of D9 mutants because of the small number of D9/+ females in the F2 generation, which made it difficult to maintain the numbers of D9/+ and D9/D9 necessary for analyses in subsequent generations. We are currently investigating the reason for the decrease in the number of D9/+ females. However, as the present study aimed to evaluate the effects of rad50 frameshift mutation with premature translation stop on tumorigenesis, lifespan, A-T, etc., we discussed the results obtained from the analyses of D2/+. 

Accordingly, we have added the descriptions related to the D9/+ and D9/D9 phenotypes to clarify why the data related to D9 has not been presented and improve overall understandability. Please see the following sentences that have been added to different parts of the revised manuscript;

In the Results section:

(p.15, line 344-346) “rad50D9/+ medaka were subjected to survival time measurement and histological analyses to determine whether genome-editing frameshift mutations without putative premature stop codons affected lifespans and tumorigenesis.” 

(p.15, line 347-350) “However, these homozygotes could not be analyzed in detail because of their smaller sample sizes. In-crosses of rad50D9/+ medaka produced only three rad50D9/D9 medaka with 63, 68, and 70 weeks or more lifespans, which were close to those of wildtype medaka, suggesting that the 9-basepair deletion did not affect the lifespan of medaka.” 

In the Discussion section:

(p.19, line 425-428) “On the contrary, no abnormalities in swimming and lifespan were observed in rad50D9/D9 medaka in a normal rearing condition, although those with a small sample size were preliminarily analyzed, suggesting that the 2 bp deletion is more harmful than the 9 bp deletion in terms of swimming abilities and lifespan of medaka.”

(p.19, line 428-431) “Given the short survival time and small sample sizes of the homozygotes, the effects of rad50 frameshift mutation with premature translation stop on tumorigenesis, lifespan, A-T, etc., were mainly discussed based on the comparison between wildtype and rad50D2/+ medaka.” 

Comment 2: As the D2/+ clearly shows a phenotype it would be interesting to see what the phenotypes of the homozygous D2/D2 fish are (despite shortened lifespan). Reviewer #1 thinks that brain phenotype would be particularly important to show.

Response: Thank you for this insightful suggestion. We agree that analyzing the brain phenotype of the D2/D2 fish could have provided better insights into the conclusions of this study. However, we could not perform the histology of the brains of D2/D2 fish because of their shorter lifespan, for which we could not restore the samples of these fish for histological analyses. Nevertheless, we analyzed the rheotaxis ability based on the swimming abilities of these fish. Following the editor's comment, we have added the following descriptions in the Results and Discussion sections as follows: 

In the Results section:

(p.15, line 342-343) “The swimming was unstable even in rearing water without water flow.” 

(p.15, line 343-344) “The reduced lifespan of rad50D2/D2 homozygotes made it difficult to maintain the samples before analyses.” 

In the Discussion section:

 (p.19, line 422-425) “All rad50D2/D2 medaka were unstable in swimming even under normal conditions without water flow, suggesting the possibility that the rheotaxis ability of the homozygous mutants was lower than that of heterozygotes.”

Comment 3: All minor concerns raised by Reviewer #1 should be addressed by making appropriate changes to the text.

Response: Thank you for your suggestion. We have addressed all comments raised by reviewer #1 and revised the manuscript accordingly. 

Thank you again for your insightful comments on our manuscript. We believe that the revised manuscript is now suitable for publication.

 

Reviewer #1

Major comment: The authors generated two mutants, the targeted frameshift deletion variant D2 and a non-targeted in-frame deletion variant D9 (the latter one was also followed as potentially mild protein variant). While homozygotes for the D2 variant had a mean survival of only six weeks, it is not clear what the fate of D9 homozygotes was. But this would be important to judge about the role of D9 in rad50 function. Also, D9 mutants seem to have not been assessed in all assays (e.g. Table 2). Furthermore, D2 homozygotes were apparently not assessed for brain phenotypes. The manuscript might benefit from these additional informations and from a side-by-side comparison of results for heterozygotes and homozygotes for both mutants.

Response: Thank you for these insightful suggestions. We agree that our manuscript could have benefited from a side-by-side comparison of heterozygous and homozygous D2 and D9 mutants. However, our study did not allow side-by-side comparisons because of the short life span and small body size of D2/D2. This made it difficult to use the samples for histological analyses while avoiding tissue deterioration. Nevertheless, we observed erratic swimming abilities in D2/D2 mutants under normal rearing conditions. Therefore, we have added the following texts to the Results and Discussion sections,

In the Results section:

(p.15, line 342-343) “The swimming was unstable even in rearing water without water flow.” 

(p.15, line 343-344) “The reduced lifespan of rad50D2/D2 homozygotes made it difficult to maintain the samples before analyses.” 

In the Discussion section:

(p.19, line 422-425) “All rad50D2/D2 medaka were unstable in swimming even under normal conditions without water flow, suggesting the possibility that the rheotaxis ability of the homozygous mutants was lower than that of heterozygotes.”

A comparison with D9/+ was necessary to determine whether genome-editing frameshift mutations without putative premature stop codons affected tumorigenesis and lifespans. However, due to the small number of D9/+ females in the F2 generation, it was difficult to maintain the numbers of D9/+ and D9/D9 necessary for analyses in subsequent generations. We are currently investigating the reason for the decrease in the number of D9/+ females. In this study, we obtained three D9/D9 mutants, and their lifespans were comparable to that of the wildtype. Also, no abnormalities in swimming were observed in D9/D9 in normal rearing conditions. Accordingly, we have added the following texts to the Results and Discussion sections:

In the Results section:

(p.15, line 344-346) “rad50D9/+ medaka were subjected to survival time measurement and histological analyses to determine whether genome-editing frameshift mutations without putative premature stop codons affected lifespans and tumorigenesis.” 

(p.15, line 347-350) “However, these homozygotes could not be analyzed in detail because of their smaller sample sizes. In-crosses of rad50D9/+ medaka produced only three rad50D9/D9 medaka with 63, 68, and 70 weeks or more lifespans, which were close to those of wildtype medaka, suggesting that the 9-basepair deletion did not affect the lifespan of medaka.”

In the Discussion section:

(p.19, line 425-428) “On the contrary, no abnormalities in swimming and lifespan were observed in rad50D9/D9 medaka in a normal rearing condition, although those with a small sample size were preliminarily analyzed, suggesting that the 2 bp deletion is more harmful than the 9 bp deletion in terms of swimming abilities and lifespan of medaka.”

Minor comments:

Comment 1: Abstract, line 23: The authors postulate that their model reproduced tumorigenesis. But this seems to be overstated in regard that the heterozygous medaka had no malignant tumours (which is stated in line 408). Therefore, the conclusion in the abstract would need to be reworded.

Response: We apologize for the confusion. In this study, we observed capillary hemangiomas of the retina, kidney, and nodular thyroid hyperplasia in rad50D2/+ medaka. These results substantiate the reported tumorigenic properties of RAD50, as suggested by the prevalence of RAD50 mutations in patients with cancer. Therefore, to avoid confusion, we revised the related sentence in the discussion section and retained the conclusions in the abstract as in our original submission. Please see the following changes: 

Original: 

“No malignant tumors were observed in rad50D2/+ medaka, but capillary hemangiomas of the retina, kidney, and nodular thyroid hyperplasia were noted.”

Revised (p.19, lines 432–433): 

“Capillary hemangiomas of the retina, kidney, and nodular thyroid hyperplasia were observed in rad50D2/+ medaka.”

Comment 2: Introduction, line 50: Note that the embryonic lethality in rad50 null mice is not different from the described NBSLD patients as these patients were carrying hypomorphic mutations.

Response: Thank you for highlighting this. Accordingly, we checked articles #8 and #9 and understood that the #8 citation was incorrect in line50 of the Introduction section. As noted by you, reference #8 reports the data on a patient with NBSLD with hypomorphic protein and does not match the sentence in line 50. However, reference #9 reports the data on a patient with NBSLD with a RAD50 homozygous mutation leading to a frameshift with premature translation stop and is consistent with the statement in line50 of the Introduction. Accordingly, we have changed the following text (p. 3, line 56-57) from

“which differ from an NBSLD patient with a homozygous mutation of the RAD50 gene [8, 9].”

to

“which differs from patients with NBSLD with a homozygous mutation of the RAD50 gene [9].”

In relation to this comment, we have also changed another part (p. 3, line 59-60) as follows:

“Moreover, this phenotype is not similar to that of NBSLD patients with a mutation in the RAD50 gene [8, 9].”

to

“Moreover, this phenotype is not similar to that of patients with NBSLD with a mutation in the RAD50 gene [8].”

Comment 3: Introduction, line 78: It should be clarified whether the identified mutation is a nonsense mutation or a frameshift mutation and whether it occurred in the heterozygous state.

Response: Following your comment, we have changed the following sentence (p.4, line 82-83), from

“RAD50 I505fs nonsense germline mutation”

to

“heterozygous RAD50 I505fs*5 frameshift germline mutation”

Comment 4: Introduction, line 80: Microsatellite instability is not an expected outcome of RAD50 heterozygosity. Has this colon cancer been assessed for mismatch repair deficiency?

Response: Colon and duodenal cancers have been assessed for mismatch repair deficiency to investigate whether immune checkpoint inhibitors were indicated.

Comment 5: Material and Methods, line 137: The 2 bp deletion needs to be defined by chromosome position and deleted nucleotides.

Response: Thank you for highlighting this. Following your suggestion, we have changed the following text (p. 6, line 141-144), 

“The RAD50 mutation identified in our patient with two cancers was a 2 bp deletion within exon 8, which corresponds to the putative exon 11 of medaka rad50.”

to

“The RAD50 mutation identified in our patient with two cancers was a 2 bp deletion (c.1515_1516 del AA, p.I505fs*5) within exon 8 of the gene on chromosome 5q; exon 8 of human RAD50 corresponds to the putative exon 11 of medaka rad50. The chromosomal location information of the medaka rad50 is not shown in the Ensembl genome browser.”.

Comment 6: Discussion, line 414: Note that Heikkinen et al. (Ref. 41) did not report on polymorphisms but on RAD50 truncating variants including a frameshift mutation (RAD50 687delT) that would be similar to the one in this report.

Response: Apologies for our oversight. According to your suggestion, we have changed the following text (p. 19, line 436-438), 

“However, a study in Finnish breast and ovarian cancer patients reported a significant association between RAD50 polymorphism and cancer morbidity in Finnish people.”

to

“In contrast, a study in Finnish patients with breast and ovarian cancer reported a significant association between RAD50 687delT and predisposition to cancer in Finnish people [41].”

Thank you again for your insightful comments on our paper. We believe that the revised manuscript is now suitable for publication.

Other changes made by the authors

We have added the following words (p.8, line 191-192) “the length of a fish measured from the tip of the snout to the posterior end of the last vertebra” as the definition of SL.

We have changed the following word (p.19, line 433) from “carcinogenic” to “tumorigenic”.

---

## [Editor Report · Decision Letter 1]

13 Feb 2023

A rad50 germline mutation induces tumorigenesis and ataxia-telangiectasia phenotype in a transparent medaka model

PONE-D-22-32016R1

Dear Dr. Ohtsuka,

We’re pleased to inform you that your manuscript has been judged scientifically suitable for publication and will be formally accepted for publication once it meets all outstanding technical requirements.

Kind regards,

Sebastian D. Fugmann, Ph.D.

Academic Editor

PLOS ONE
---

## [Editor Report · Acceptance letter]

13 Apr 2023

PONE-D-22-32016R1 

A *rad50* germline mutation induces tumorigenesis and ataxia-telangiectasia phenotype in a transparent medaka model 

Dear Dr. Ohtsuka:

I'm pleased to inform you that your manuscript has been deemed suitable for publication in PLOS ONE. Congratulations! Your manuscript is now with our production department. 

Kind regards, 

on behalf of

Dr. Sebastian D. Fugmann 

Academic Editor

PLOS ONE